# Position and Attitude Tracking of MAV Quadrotor Using SMC-Based Adaptive PID Controller

Aminurrashid Noordin [1,2], Mohd Ariffanan Mohd Basri [1,*] and Zaharuddin Mohamed [1]

1  School of Electrical Engineering, Faculty of Engineering, Universiti Teknologi Malaysia (UTM), Skudai 81310, Johor, Malaysia
2  Faculty of Electrical and Electronic Engineering Technology, Universiti Teknikal Malaysia Melaka, Hang Tuah Jaya, Durian Tunggal 76100, Melaka, Malaysia
*  Correspondence: ariffanan@fke.utm.my

**Abstract:** A micro air vehicle (MAV) is physically lightweight, such that even a slight perturbation could affect its attitude and position tracking. To attain better autonomous flight system performance, MAVs require good control strategies to maintain their attitude stability during translational movement. However, the available control methods nowadays have fixed gain, which is associated with the chattering phenomenon and is not robust enough. To overcome the aforementioned issues, an adaptive proportional integral derivative (PID) control scheme is proposed. An adaptive mechanism based on a second-order sliding mode control is used to tune the parameter gains of the PID controller, and chattering phenomena are reduced by a fuzzy compensator. The Lyapunov stability theorem and gradient descent approach were the basis for the automated tuning. Comparisons between the proposed scheme against SMC-STA and SMC-TanH were also made. MATLAB Simulink simulation results showed the overall favourable performance of the proposed scheme. Finally, the proposed scheme was tested on a model-based platform to prove its effectiveness in a complex real-time embedded system. Orbit and waypoint followers in the platform simulation showed satisfactory performance for the MAV in completing its trajectory with the environment and sensor models as perturbation. Both tests demonstrate the advantages of the proposed scheme, which produces better transient performance and fast convergence towards stability.

**Keywords:** micro air vehicle (MAV); quadrotor; sliding mode control; adaptive pid; fuzzy compensator; perturbation

## 1. Introduction

In recent years, researchers have shown great interest in the micro air vehicle (MAV), which is a small-scale drone with a mass of less than 0.1 kg [1] due to its reliability and safety of operation in GPS-denied workspace [2]. Since their sizes are comparatively small, MAVs are very susceptible to perturbations, such as wind gusts, and this could become a significant problem for real-time implementation. Therefore, a controller that can guarantee a successful flight is very crucial for their operation.

There are several commercialized quadrotors for education today, but this study focuses on the Parrot Mambo Minidrone, which is categorized as an MAV. The Parrot Mambo Minidrone is a 6-DOF quadrotor that includes sensors such as ultrasonic sensors, an accelerometer, gyroscopes, air pressure sensors, and down-facing cameras (optical flow). Since MATLAB provides an open architecture by Simulink Support Package for Parrot Minidrones (SSPPM), it allows for a Bluetooth low-energy connection to deploy algorithms wirelessly via a personal area network (PAN). Hence, the possibility of low-level testing control to this Parrot Mambo Minidrone is the main advantage. The researcher utilized the provided add-on hardware support offered by MATLAB based on the Aerospace Blockset developed by MIT [3].

The MAV's quadrotor fuselage is a small but underactuated and dynamically unstable system, making designing a complete controller to be such a challenging task. Due to its mass being less than 0.1 kg, the MAV's performance is easily affected by small uncertainties and external disturbances such as wind gusts and noise. Many nonlinear and linear controls have therefore been established given these circumstances. For instance, adaptive control [4–6], backstepping control [7,8], sliding mode control [9–14], optimal control [15,16] and intelligent control [17–21] have been deployed.

An adaptive sliding mode is a core of the approach used in [4], where the benefits depend on the magnitude of the gain and robustness as well as not being overestimated by small perturbations. Owing to adaptive gains, only appropriate controls are used to fulfil the task of reducing the chattering effect, thus avoiding the perturbation effect. To show the effectiveness and benefits of the suggested controller successfully, the experiment is also carried out on the commercial Parrot Rolling Spider quadrotor MAV. The researchers discussed a comprehensive, adaptive sliding mode control design approach using the radial basis function neural network (RBF NN) in [5] for the Parrot Rolling Spider quadrotor MAV. In the case of nonlinearities, the suggested system claims to have the potential to estimate uncertainties, where the NN is used as a compensator parameter. Therefore, accelerated error reduction can be accomplished in the closed-loop control system.

By focusing on the MAV quadrotor, [9] simulated the uses of a hierarchical perturbation compensator (HPC) to improve the sliding mode controller. The three compensators, feed-forward perturbation compensator, feed-back perturbation compensator and sliding mode perturbation compensator, were built to reduce the perturbation hierarchically. Later on, [10,22] studied a super-twisting second order sliding mode controller based on a modified non-singular terminal sliding surface and successfully implemented it on the Parrot Rolling Spider quadrotor MAV. The control algorithm includes a new exponent term for a nonlinear switching surface that bypasses the singularity problem. Hence, the system achieved stability conditions with chattering phenomenon reduction to ensure the system's robustness by perturbation rejection.

A conventional PID control technique was compared with optimal control in [15] to achieve stability at the hovering condition of the Parrot Rolling Spider quadrotor MAV. This paper introduces optimal control by an LQR controller that can ignore the difficulty in tuning the PID control parameters. Moreover, instead of six PID controllers, only one LQR controller is used to better track the system with minimum settling time in the simulation. The PI-PD and Fuzzy PI-PD controller were implemented on the Parrot Mambo Minidrone quadrotor MAV as reported in [23]. The FLC settings of $3 \times 3$ rules were based on triangle membership function as the input and singletons as the output. Finally, the center of sets was chosen for its defuzzification method to provide robustness against nonlinearities. Ref. [21] also proposed a PID control approach, which is based on an adaptive fuzzy PID-UAV attitude controller. It has been established that the control system has the advantages of high precision and ease of implementation. However, the fuzzy controller has the advantages of minimal overshoot and transient response, as well as the ability to achieve precise and rapid behavior control.

A fuzzy hybrid scheme that comprises fuzzy logic control to fine-tune the regulation pole-placement gains was proposed in [19]. For the stability of tri-motor UAV, this model reference adaptive control (MRAC) is utilized, which shows this algorithm has better transient performance with zero steady-state and fast convergence towards stability. To overcome the complexity of a coupled nonlinear model of a fixed-wing UAV system, [20] proposed a control algorithm combining fuzzy adaptive control and sliding mode variable structure control. The stability of the controllers was verified using the Lyapunov stability theorem. This method has a fast response speed, small steady-state error and strong robustness. It is recommended for complicated, nonlinear, strongly coupled and multiple uncertain models, such as the fixed-wing and multiple-rotor UAV model.

Ref. [24] proposed an auto-tuning PID using a sliding mode approach for a DC servomotor using FPGA, which showed better tracking responses after control parameter

training. In [25], an adaptative PID control using sliding mode control has been proposed on a quadrotor UAV's attitude, and the position control stabilization was compared to PID and SMC. In addition, a fuzzy compensator was included to reduce the chattering phenomenon. For the simulation part, a wind force disturbance was applied to show the robustness of the proposed control scheme with up to 100% changes in parameter uncertainties.

The motivation of this research is to develop an effective flight control system for a small micro aerial vehicle, which is less than 0.1 kg in weight. This is due to the MAV being relatively small and lightweight, and therefore its attitude and position tracking can be easily affected by a slight perturbation. Therefore, to attain a better-performing autonomous flight system, the MAV requires good control strategies to maintain attitude stability during translational movement. The main contribution of this paper is the implementation of an adaptive PID controller (APID) scheme using sliding mode control on the Parrot Mambo Minidrone quadrotor MAV, which has a mass of less than 0.1 kg. Ref. [25] implemented this scheme on a UAV, while [24] applied it to a DC motor. Nevertheless, in this paper, the proposed APIDC controller performances were compared to SMC with super twisting and SMC with a TanH function, whereas it was PID and SMC in [25]. By using sliding mode control as the adaptive mechanism, this approach may overcome the re-tuning gains of a proportional-integral-derivative controller's manual controller. In addition, the chattering effect by sliding manifolds is eliminated using a fuzzy compensator. To demonstrate the advantages of the proposed flight control system, simulation and real-time experiments were conducted with a few adjustments to the flight control system in SSPPM. Furthermore, this paper also considers external disturbances from the environment model and noise from the sensors model. In conclusion, the APIDC system can produce better transient performance and fast convergence towards stability.

The organization of this manuscript is as follows. Section II describes the dynamics model of the MAV quadrotor. Section III shows the design of the APIDC for the altitude and attitude control of the quadrotor. Section IV presents the simulation and experimental results of the APIDC system, and finally, Section V presents the conclusion of the research work.

## 2. Modelling of MAV

The Parrot Mambo Minidrone is a quadrotor that belongs to the micro aerial vehicle (MAV) category, which comprises four symmetrical arranged rotors and is independently mounted on a rigid fuselage, as illustrated in Figure 1. Rotors (1 and 3) rotate clockwise, while rotors (2 and 4) rotate counterclockwise.

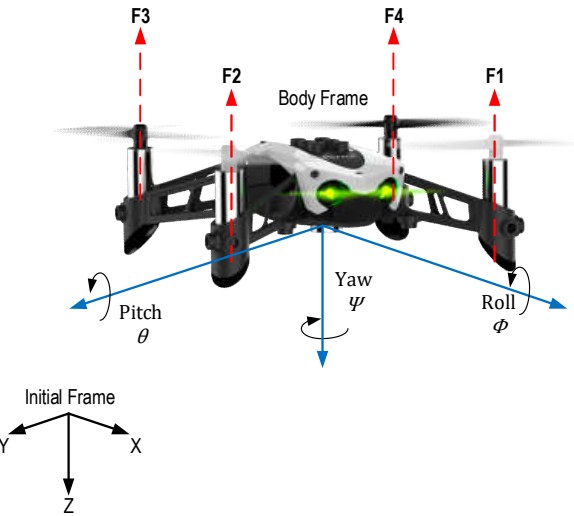

**Figure 1.** Parrot Mambo Minidrone MAV.

The mathematical model of this MAV is standardized in 6 degrees of freedom (DOF) $[x, y, z, \phi, \theta, \psi]^T$, as referred to by researchers of UAV systems. Thus, the model may be split into two coordinate subsystems, where the vector $[x, y, z]^T$ describes the absolute positions, and the vector $[\phi, \theta, \psi]^T$ describes the orientation.

From the general Newton–Euler formulation, quadrotor translational dynamics are described as:

$$
\begin{aligned}
m\ddot{x} &= -U_1(sin(\psi)sin(\phi) + cos(\psi)sin(\theta)cos(\phi)) \\
m\ddot{y} &= -U_1(-cos(\psi)sin(\phi) + sin(\psi)sin(\theta)cos(\phi)) \\
m\ddot{z} &= mg - U_1(cos(\theta)cos(\phi))
\end{aligned}
\tag{1}
$$

where $m$ is the mass, $g$ is the gravitational coefficient, and $U_1$ is the total thrust force. Meanwhile, the rotational dynamics are described as:

$$
\begin{aligned}
I_{xx}\ddot{\phi} &= \left(I_{yy} - I_{zz}\right)\dot{\psi}\dot{\theta} - (J_r\Omega_d)\dot{\theta} + lU_2 \\
I_{yy}\ddot{\theta} &= (I_{zz} - I_{xx})\dot{\psi}\dot{\phi} + (J_r\Omega_d)\dot{\phi} + lU_3 \\
I_{zz}\ddot{\psi} &= (I_{xx} - I_{yy})\dot{\theta}\dot{\phi} + U_4
\end{aligned}
\tag{2}
$$

where $U_2$, $U_3$ and $U_4$ are the total torque for the roll, pitch, and yaw, respectively; $l$ is the lateral arm length; $I_{xx}$, $I_{yy}$, and $I_{zz}$ are the moments of inertia for the quadrotor; $J_r$ is the rotor inertia, and $\Omega_d$ is the total rotor speed generated from the torque related to the control inputs as:

$$
\begin{aligned}
u_1 &= b\left(\Omega_1^2 + \Omega_2^2 + \Omega_3^2 + \Omega_4^2\right) \\
u_2 &= b\left(\Omega_1^2 - \Omega_2^2 - \Omega_3^2 + \Omega_4^2\right) \\
u_3 &= b\left(\Omega_1^2 + \Omega_2^2 - \Omega_3^2 - \Omega_4^2\right) \\
u_4 &= d\left(-\Omega_1^2 + \Omega_2^2 - \Omega_3^2 + \Omega_4^2\right) \\
\Omega_d &= -\Omega_1 + \Omega_2 - \Omega_3 + \Omega_4
\end{aligned}
\tag{3}
$$

where $\Omega_i (i = 1, 2, 3, 4)$ is the rotor speed, and $b$ and $d$ are the thrust and drag coefficients, respectively. Finally, the Parrot Mambo Minidrone parameters are listed in Table 1.

**Table 1.** Parrot Mambo Model Physical Parameters [2].

| Specification | Parameter | Unit | Value |
|---|---|---|---|
| Quadrotor mass | $m$ | kg | 0.0630 |
| Lateral moment arm | $l$ | m | 0.0624 |
| Thrust coefficient | $b$ | Ns$^2$ | 0.0107 |
| Drag coefficient | $d$ | Nms$^2$ | $0.7826400 \times 10^{-3}$ |
| Rolling moment of inertia | $I_{xx}$ | kgm$^2$ | $0.0582857 \times 10^{-3}$ |
| Pitching moment of inertia | $I_{yy}$ | kgm$^2$ | $0.0716914 \times 10^{-3}$ |
| Yawing moment of inertia | $I_{zz}$ | kgm$^2$ | $0.1000000 \times 10^{-3}$ |
| Rotor moment of inertia | $J_r$ | kgm$^2$ | $0.1021 \times 10^{-6}$ |

*2.1. Quadrotor State Space Representation*

The dynamics provided in (1) and (2) are expressed as a conventional structure of second-order state-space equations:

$$
\ddot{x} = f(x) + g(x)u
\tag{4}
$$

To carry out a systematic control system design approach while maintaining simple notation, the state vector is expressed as:

$$
x = \begin{bmatrix} x_1 & x_2 & x_3 & x_4 & x_5 & x_6 \end{bmatrix}^T = \begin{bmatrix} x & y & z & \phi & \theta & \psi \end{bmatrix}^T
\tag{5}
$$

And the input vector is $u = [u_1 \; u_2 \; u_3 \; u_4]^T$. The nonlinear function $f(x)$ and $g(x)$ can be rewritten as:

$$f(x) = \begin{bmatrix} f_1(x) \\ f_2(x) \\ f_3(x) \\ f_4(x) \\ f_5(x) \\ f_6(x) \end{bmatrix} = \begin{bmatrix} 0 \\ 0 \\ g \\ a_1\dot{\psi}\dot{\theta} - a_2\Omega_d\dot{\theta} \\ a_3\dot{\psi}\dot{\phi} + a_4\Omega_d\dot{\phi} \\ a_5\dot{\theta}\dot{\phi} \end{bmatrix} \tag{6}$$

$$g(x) = \begin{bmatrix} g_1(x) & 0 & 0 & 0 \\ g_2(x) & 0 & 0 & 0 \\ g_3(x) & 0 & 0 & 0 \\ 0 & g_4(x) & 0 & 0 \\ 0 & 0 & g_5(x) & 0 \\ 0 & 0 & 0 & g_6(x) \end{bmatrix} = \begin{bmatrix} -\frac{1}{m}u_x & 0 & 0 & 0 \\ -\frac{1}{m}u_y & 0 & 0 & 0 \\ -\frac{1}{m}u_z & 0 & 0 & 0 \\ 0 & b_1 & 0 & 0 \\ 0 & 0 & b_2 & 0 \\ 0 & 0 & 0 & b_3 \end{bmatrix} \tag{7}$$

where

$$a_1 = \frac{I_{yy} - I_{zz}}{I_{xx}}, \; a_2 = \frac{J_r}{I_{xx}}, \; a_3 = \frac{I_{zz} - I_{xx}}{I_{yy}}$$
$$a_4 = \frac{J_r}{I_{yy}}, \; a_5 = \frac{I_{xx} - I_{yy}}{I_{zz}}, \; b_1 = \frac{1}{I_{xx}} \tag{8}$$
$$b_2 = \frac{1}{I_{yy}}, \; b_3 = \frac{1}{I_{zz}}$$

and

$$u_x = S\psi S\phi + C\psi S\theta C\phi$$
$$u_y = -C\psi S\phi + S\psi S\theta C\phi \tag{9}$$
$$u_z = C\theta C\phi$$

### 2.2. Position Control

$U_1$ cannot be used to control the quadrotor trajectory's $x$ and $y$ positions. However, the roll and pitch angles can be used to control the $x$ and $y$ positions. Since the quadrotor's operation relies on the hover position, minimal angle values for the roll and pitch angles are required. For a simplified translation dynamic, the small-angle assumptions $(S\phi_d \cong \phi_d, \; S\theta_d \cong \theta_d, \; C\phi_d = C\theta_d = 1)$ are used in (1) as follows:

$$\ddot{x} = -\frac{1}{m}(\phi_d S\psi + \theta_d C\psi)U_1$$
$$\ddot{y} = -\frac{1}{m}(-\phi_d C\psi + \theta_d S\psi)U_1 \tag{10}$$

In a matrix form:

$$\begin{bmatrix} \ddot{x} \\ \ddot{y} \end{bmatrix} = \frac{U_1}{m} \begin{bmatrix} -S\psi & -C\psi \\ C\psi & -S\psi \end{bmatrix} \begin{bmatrix} \phi_d \\ \theta_d \end{bmatrix} \tag{11}$$

Therefore, by inverting (11), the desired angle of the roll, $\phi_d$ and the pitch, $\theta_d$ may be calculated as follows:

$$\phi_d = \frac{m}{U_1}(S\psi_d\ddot{x} - C\psi_d\ddot{y})$$
$$\theta_d = \frac{m}{U_1}(C\psi_d\ddot{x} + S\psi_d\ddot{y}) \tag{12}$$

### 3. Flight Control Design

The flight control is designed by implementing the following algorithms:

i.      The tracking error is set as $e = x_d - x$;

ii.     The sliding surface is chosen as $s = \dot{e} + k_1 e + k_2 \int e dt$;

iii.     The PID controller, $u_{pid} = \hat{k}_p e + \hat{k}_i \int e dt + \hat{k}_d \dot{e}$ where parameter gains $\hat{k}_p$, $\hat{k}_i$ *and* $\hat{k}_d$ are updated by

      a.    $\dot{\hat{k}}_p = \beta_p s e$;

      b.    $\dot{\hat{k}}_i = \beta_i s \int e dt$;

      c.    $\dot{\hat{k}}_d = \beta_d s \dot{e}$;

iv.      The fuzzy compensator $u_{fzc} = \hat{r}(w_1 - w_3)$ where the parameter $\hat{r}$ is estimated by $-\eta_r s(w_1 - w_3)$;

v.      The control law is given by $u = u_{pid} + u_{fzc}$.

The control objective of the system is to find a control law so that the $x$ can track the desired $x_d$ closely; thus, the tracking error can be defined as:

$$e = x_d - x \tag{13}$$

Consider that an optimum controller exists if all the variables in (4) are known and can be described as:

$$u^* = g^{-1}\left(-f + \ddot{x}_d + k_1\dot{e} + k_2 e\right) \tag{14}$$

Hence, to meet the Hurwitz criterion, $k_1$ and $k_2$ are selected as non-zero positive constants., which implies $\lim_{t\to\infty} e = 0$ for any initial starting conditions. Substituting (14) into (4) yields:

$$\ddot{e} + k_1\dot{e} + k_2 e = 0 \tag{15}$$

The ideal controller u* in (14) cannot be precisely determined since the system dynamics are typically unknown in actual applications. Nevertheless, this problem may be solved by using a sliding mode controller. The nominal model (4) may be reformulated as follows before building the requisite controller:

$$\ddot{x} = f_n(x) + g_n(x)u \tag{16}$$

where $f_n$ and $g_n$ represent the nominal behaviour of $f$ and $g$, respectively. If uncertainties and external disturbances are considered, (16) can be further modified as:

$$\ddot{x} = (f_n + \Delta f) + (g_n + \Delta g)u + d = f_n + g_n u + w \tag{17}$$

where $d$ is the external disturbance, i.e., wind, $\Delta f$ and $\Delta g$ are system uncertainties, $w$ is lumped of uncertainties and defined as $w = \Delta f + \Delta g u + d$, with the assumption $|w| \le W$, where $W$ is a positive constant. The main idea behind sliding mode control is to make sure the system satisfies the necessary condition and is certain to have a sliding condition. Therefore, the chosen characteristics of a sliding surface in this paper are as follows:

$$s = \dot{e} + k_1 e + k_2 \int e \, dt \tag{18}$$

where $k_1$ and $k_2$ are positive constants. The sliding-mode control law is outlined in [24,25] as:

$$u_{sc} = u_{eq} + u_{ht} \tag{19}$$

where $u_{eq}$ is the equivalent controller and is defined as:

$$u_{eq} = g_n^{-1}\left(-f + \ddot{x}_d + k_1\dot{e} + k_2 e\right) \tag{20}$$

The hitting controller, $u_{ht}$ is designed to guarantee system stability and is described as:

$$u_{ht} = \frac{W}{g_n}sign(s), \; where \; sign(s)\begin{cases} +1 \; if \; s < 0 \\ -1 \; if \; s > 0 \end{cases} \tag{21}$$

The derivative of (18) gives:

$$\dot{s} = \ddot{e} + k_1\dot{e} + k_2 e \tag{22}$$

Inserting (19)–(21) into (17) yields:

$$\ddot{e} + k_1\dot{e} + k_2 e = -w - Wsign(s) = \dot{s} \tag{23}$$

For stability, a Lyapunov function is considered as follows:

$$V_1 = \tfrac{1}{2}s^2 \tag{24}$$

Differentiating (24) to time gives:

$$\dot{V}_1 = s\dot{s} \tag{25}$$

For stability $\dot{V}_1 \leq 0$, replacing (23) into (25) yields:

$$\dot{V}_1 = -(W - |w|)|s| \tag{26}$$

In short, system stability can be achieved using the sliding mode control principles based on the Lyapunov theorem. A bigger control gain, $W$, on the other hand, will generate chattering. Moreover, the switching function is not easy to implement due to physical limitations on the rotor used by an MAV [24,25].

The block diagram for the quadrotor altitude and attitude control systems in an adaptive PID system is shown in Figure 2. A PID controller and a fuzzy logic compensator are included in the proposed approach as:

$$u = u_{pid} + u_{fzc} \tag{27}$$

here, $u_{pid}$ is the PID controller used to approximate the optimum controller, u*, and $u_{fzc}$ is the fuzzy logic compensators designed to reduce the remaining approximation error between both controllers [24,25].

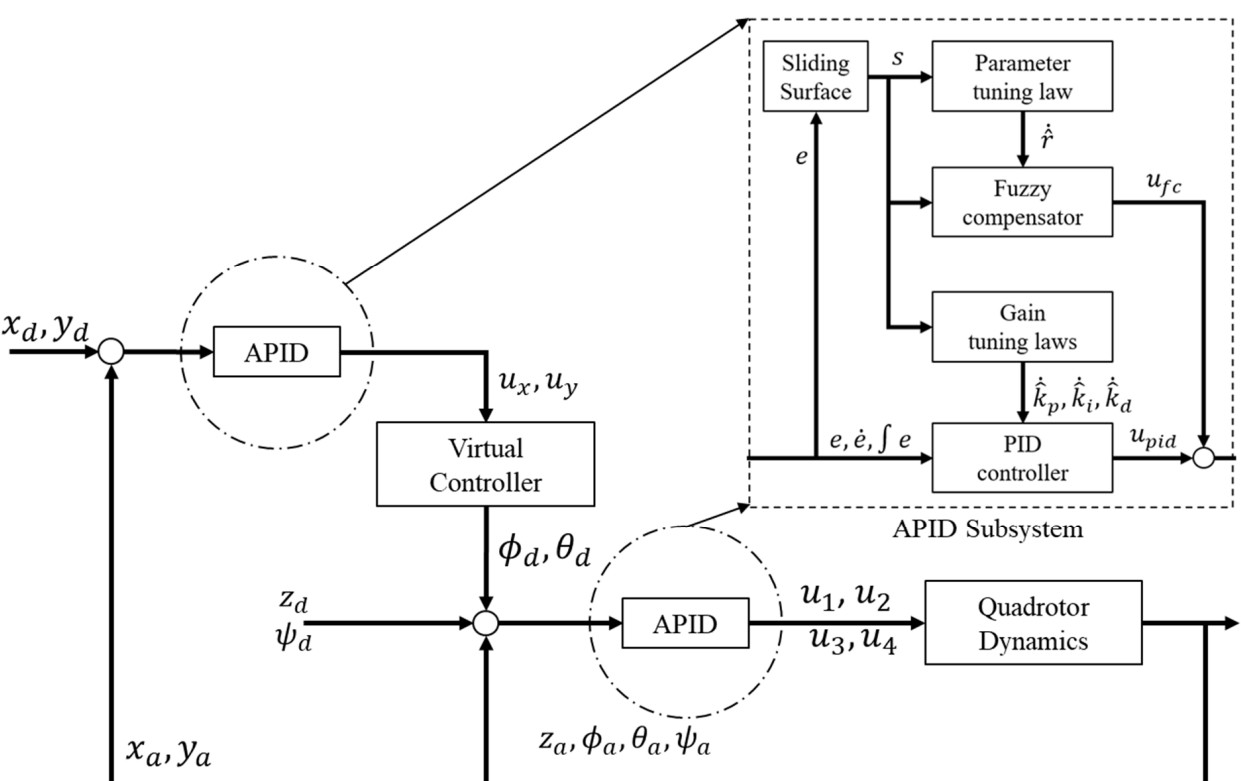

**Figure 2.** Block diagram of an adaptive PID controller for a quadrotor UAV.

*3.1. Pid Controller Design*

The conventional PID controller is described as:

$$u_{pid} = \hat{k}_p e + \hat{k}_i \int edt + \hat{k}_d \dot{e} \tag{28}$$

where $\hat{k}_p$, $\hat{k}_i$, and $\hat{k}_d$ are the value of proportional, integral, and derivative gain, respectively. This paper proposed an adaptive mechanism to online determine these controller gains. For example, with a derivative of both sides of (22) and using (4), it can be observed that:

$$\dot{s} = \ddot{e} + k_1\dot{e} + k_2 e = -f - gu + A_d \tag{29}$$

where, $A_d = \ddot{x}_d + k_1\dot{e} + k_2 e$, substitute (27) into (29) and multiply by $s$ gives,

$$s\dot{s} = -s\left[g\left(u_{pid} + u_{fzc}\right) + f - A_d\right] \tag{30}$$

Using the gradient approach and the chain rule, $\hat{k}_p$, $\hat{k}_i$, and $\hat{k}_d$ gains are updated using the principles given in [24,25]:

$$\begin{aligned}
\dot{\hat{k}}_p &= s\beta_p sign(g)e = \beta_p se \\
\dot{\hat{k}}_i &= s\beta_i sign(g) \int edt = \beta_i s \int edt \\
\dot{\hat{k}}_d &= s\beta_d sign(g)\dot{e} = \beta_d s\dot{e}
\end{aligned} \tag{31}$$

where $\beta_p$, $\beta_i$, and $\beta_d$ represent the positive learning rates of $\dot{\hat{k}}_p$, $\dot{\hat{k}}_i$, and $\dot{\hat{k}}_d$, respectively. Additionally, the design method requires the g value, which may be easily obtained from the controlled system's physical characteristics [24,25].

*3.2. Design of the Fuzzy Compensator*

Three ambiguous rules are presented for the compensator as described when P is positive, Z is zero, and N is negative in the expression below:

$$\begin{aligned}
&Rule\ 1: \ If\ s\ is\ P,\ the\ u_{fzc}\ is\ P \\
&Rule\ 2: \ If\ s\ is\ Z,\ the\ u_{fzc}\ is\ Z \\
&Rule\ 3: \ If\ s\ is\ N,\ the\ u_{fzc}\ is\ N
\end{aligned} \tag{32}$$

where the input and output membership functions, as depicted in Figure 3a,b, are defined using triangular and singleton features. Finally, the defuzzification is achieved using the center-of-gravity approach as described below:

$$u_{fzc} = \frac{\sum_{i=1}^{3} r_i w_i}{\sum_{i=1}^{3} w_i} = r_1 w_1 + r_2 w_2 + r_3 w_3 \tag{33}$$

where the firing strength, $w_i$ ($i$ = 1, 2, 3), is bounded as ($0 \leq w_i \leq 1$) and the sum of $w_i$ is exclusive to the case of the triangle membership function, being not more than one. Let us reduce the strain on the calculations by $r_1 = \hat{r}$, $r_2 = 0$, $and\ r_3 = -\hat{r}$. As a result, only one of the four scenarios will occur for any value of input $s$, as mentioned in [24,25].

Case 1: Only rule 1 is triggered ($s > s_a$, $w_1 = 1$, $w_2 = w_3 = 0$)

$$u_{fzc} = r_1 = \hat{r} \tag{34}$$

Case 2: Rules 1 and 2 are triggered simultaneously ($0 < s \leq s_a$, $0 < w_1$, $w_2 \leq 1, w_3 = 0$)

$$u_{fzc} = r_1 w_1 = \hat{r} w_1 \tag{35}$$

Case 3: Rules 2 and 3 are triggered simultaneously ($s_b < s \leq 0$, $w_1 = 0$, $0 < w_2$, $w_3 \leq 1$)

$$u_{fzc} = r_3 w_3 = -\hat{r} w_3 \tag{36}$$

Case 4: Only rule 3 is triggered ($s \leq s_b$, $w_1 = w_2 = 0$, $w_3 = 1$)

$$u_{fzc} = r_3 = -\hat{r} \tag{37}$$

Then, (34)–(37) can be reproduced as:

$$u_{fzc} = \hat{r}(w_1 - w_3) \tag{38}$$

Furthermore, it can be seen from [24,25]

$$s(w_1 - w_3) = |s||(w_1 - w_3)| \geq 0 \tag{39}$$

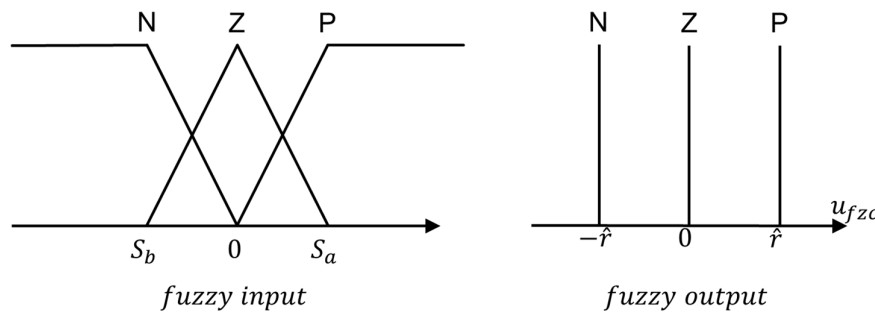

**Figure 3.** The fuzzy membership function. P is positive, Z is zero, and N is negative.

### 3.3. Stability Analysis

By substituting (27) into (4), it is revealed that

$$\ddot{x} = f + g\left(u_{pid} + u_{fzc}\right) \tag{40}$$

The error equation regulating the system can also be found by a simple manipulation of (14), (18), and (40), as shown below:

$$\ddot{e} + k_1 \dot{e} + k_2 e = g\left(u^* - u_{pid} - u_{fzc}\right) = \dot{s} \tag{41}$$

The optimal controller can be rewritten if an approximation error occurs as:

$$u^* = u_{pid}\left(\hat{k}_p, \hat{k}_i, \hat{k}_d\right) + \varepsilon \tag{42}$$

where the approximation error, $\varepsilon$, is limited to $0 \leq |\varepsilon| \leq E$, as $E$ is a positive constant. Consequently, (41) is rewritable as:

$$\dot{s} = g\left(\varepsilon - u_{fzc}\right) \tag{43}$$

Next, Lyapunov's function is chosen as:

$$V_2 = \tfrac{1}{2}s^2 \tag{44}$$

Differentiating (44) with respecting to time and applying (38) and (43) gives:

$$\dot{V}_2 = -g|s||w_1 - w_3|\left(\hat{r} - \tfrac{|\varepsilon|}{|w_1 - w_3|}\right) \tag{45}$$

If the inequality that occurs in

$$\hat{r} > \frac{|\varepsilon|}{|w_1 - w_3|} \tag{46}$$

holds, then the sliding condition $\dot{V}_2 \leq 0$ can be met. For practical applications, the value of $\hat{r}$ cannot be determined in advance because of unidentified lumped uncertainties. According to (38), to attain and match the smallest sliding value, the optimal, $r^*$ value is set as follows:

$$r^* = \frac{|\varepsilon|}{|w_1 - w_3|} + \kappa \tag{47}$$

where $\kappa$ stands for a positive constant. This research thus employs a simple adaptive algorithm to determine an optimal value, $r^*$. Hence the definition of the estimated error is:

$$\widetilde{r} = r^* - \hat{r} \tag{48}$$

where $\hat{r}$ represents the estimated optimal value of $r^*$. A new Lyapunov function is then defined as:

$$V_3 = \tfrac{1}{2}s^2 + \tfrac{g}{2\eta_r}\widetilde{r}^2 \tag{49}$$

where a constant learning rate $\eta_r > 0$. Differentiating (49) and substituting using (38), (43), and (47) yields:

$$\dot{V}_3 = g|s||\varepsilon| - \widetilde{r}g\left[s(w_1 - w_3) + \frac{\dot{\widetilde{r}}}{\eta_r}\right] - r^*sg(w_1 - w_3) \tag{50}$$

We choose the parameter tuning law as:

$$\dot{\widetilde{r}} = -\dot{\hat{r}} = -\eta_r s(w_1 - w_3) \tag{51}$$

By using (47), (50) becomes:

$$\dot{V}_3 = -g\kappa|s||w_1 - w_3| \leq 0 \tag{52}$$

Since $\dot{V}_3$ is negative semidefinite, that is $V_3(s, \widetilde{r}, t) \leq V_3(s, \widetilde{r}, 0)$, it implies that $s$ and $\widetilde{r}$ are bounded. Let function $\Omega(\tau) \equiv g\kappa|s||w_1 - w_3| \leq -\dot{V}_3$; integrating $\Omega(t)$ yields:

$$\int_0^t \Omega(\tau)d\tau \leq V_3(s, \widetilde{r}, 0) - V_3(s, \widetilde{r}, t) \tag{53}$$

Because $V_3(s, \widetilde{r}, 0)$ is bounded, and $V_3(s, \widetilde{r}, t)$ is non-increasing and bounded, the following result can be obtained:

$$\lim_{t \to \infty} \int_0^t \Omega(\tau)d\tau < \infty \tag{54}$$

Moreover, since $\dot{\Omega}(t)$ is bounded by Barbalat's Lemma [24,25], $\lim_{t \to \infty}\Omega(t) = 0$, $s \to 0$ as $t \to \infty$. Therefore, the proposed APIDC system's stability can be guaranteed.

In short, if the $\beta_p$, $\beta_i$, and $\beta_d$ learning rate or initial $\hat{k}_p$, $\hat{k}_i$, and $\hat{k}_d$ of PID gains are not correctly chosen, the system's state will diverge. Therefore, the learning rate can be adjusted manually or with the optimization method. The fuzzy compensator in Equation (38) is crucial in providing additional input for a state reversal.

## 4. Simulation Result

The simulation was set up on MATLAB Simulink based on dynamics Equations (1) and (2) and with the Parrot Mambo Minidrone parameters as tabulated in Table 1. The above-proposed controller, APIDC, was compared to a sliding mode control with TanH function (SMC TanH) [26] and a sliding mode control with a super twisting algorithm (SMCSTA) [27].

The states were initially set as $\zeta = [0,0,0,0,0,1]$ for $x, y, z, \phi, \theta, \psi$, respectively. The quadrotor was set to hover at 1 meter, and the desired position of $x$ and $y$ were set as follows:

$$x_d = sin\left(\frac{2\pi t}{60}\right)$$
$$y_d = cos\left(\frac{2\pi t}{60}\right)$$

(55)

The desired $\psi$ was set as 0 rad, while the desired $\phi$ and $\theta$ were produced by the virtual controller as stated in Equation (12). The simulation ran about 60 s.

For the super twisting sliding mode (STA), the parameters were set as follows: $\lambda_i = 1.2$, $k_{1i} = 1$, $k_{2i} = 0.002$, for $(i = z, \phi, \theta, \psi)$, $\lambda_j = 0.7$, $k_{1j} = 0.5$, and $k_{2j} = 0.001$ for $(j = x, y)$.

For the sliding mode with TanH (SMC TanH), the parameters were set as follows: $\lambda_i = 1.5$, $\eta_i = 2$, $k_i = 1$, and $\epsilon_i = 0.01$, for $(i = \phi, \theta, \psi)$, $\lambda_{(z,x,y)} = (1.2, 1.44, 1.44)$, $\eta_{z,x,y} = (3, 0.36, 0.36)$, $k_{z,x,y} = (0.01, 0.018, 0.018)$, and finally $\epsilon_{z,x,y} = (0.01, 0.018, 0.018)$.

For the adaptive PID (APID), the parameters were set as follows: $\beta_{p_j} = 1$, $\beta_{ij} = 0$, $\beta_{dj} = 1$, $k_{1j} = 1$, $k_{2j} = 0.1$, $n_{fzj} = 0.01$, for $(j = \phi, \theta, \psi)$. $\beta_{p_k} = 1$, $\beta_{ik} = 0$, $\beta_{dk} = 1$, $k_{1k} = 1$, $k_{2k} = 0.01$, and $n_{fzj} = 0.1$ for $(k = z, x, y)$.

These parameters are set to achieve better performance by comparing the integral square error (ISE) index performances of the above controllers without and with perturbation during simulation.

### 4.1. Circle Trajectory without Perturbation

The states' response is shown in Figure 4a. All states can follow the desired trajectory with minimum errors. Based on Figure 4b, APID shows a more significant response to follow the desired reference with satisfaction of integral square error (ISE) performance compared to another controller, as stated in Table 2. Figure 5 shows the input response towards the system to achieve stabilization where STA produces little chattering on input $U_2$, in contrast to SMC with TahH, which produces chattering at inputs $U_1$, $U_2$, and $U_3$ but is clean with the APID. Hence, it clearly shows the most excellent performance of APID compared to another controller.

**Table 2.** ISE index performances without and with perturbation.

| | Without perturbation | | |
|---|---|---|---|
| | **STA** | **SMC TanH** | **APID** |
| $x$ | 0.08982 | 1.1560 | 0.596900 |
| $y$ | 0.01400 | 0.9769 | 0.008548 |
| $z$ | 1.32090 | 0.6067 | 0.953600 |
| $\phi$ | $2.0080 \times 10^{-3}$ | $4.534 \times 10^{-5}$ | $3.635 \times 10^{-7}$ |
| $\theta$ | $3.0520 \times 10^{-3}$ | $7.990 \times 10^{-5}$ | $4.972 \times 10^{-3}$ |
| $\psi$ | $0.6807 \times 10^{-3}$ | $8.362 \times 10^{-9}$ | $3.646 \times 10^{-14}$ |
| | With perturbation | | |
| | **STA** | **SMC TanH** | **APID** |
| $x$ | 0.096550 | 2.0950 | 0.58480 |
| $y$ | 0.017780 | 1.5850 | 0.01069 |
| $z$ | 1.329000 | 0.6069 | 0.95470 |
| $\phi$ | $2.1250 \times 10^{-3}$ | $0.268 \times 10^{-3}$ | $4.759 \times 10^{-5}$ |
| $\theta$ | $3.4190 \times 10^{-3}$ | $0.319 \times 10^{-3}$ | $4.950 \times 10^{-3}$ |
| $\psi$ | $0.6821 \times 10^{-3}$ | $2.385 \times 10^{-7}$ | $5.385 \times 10^{-7}$ |

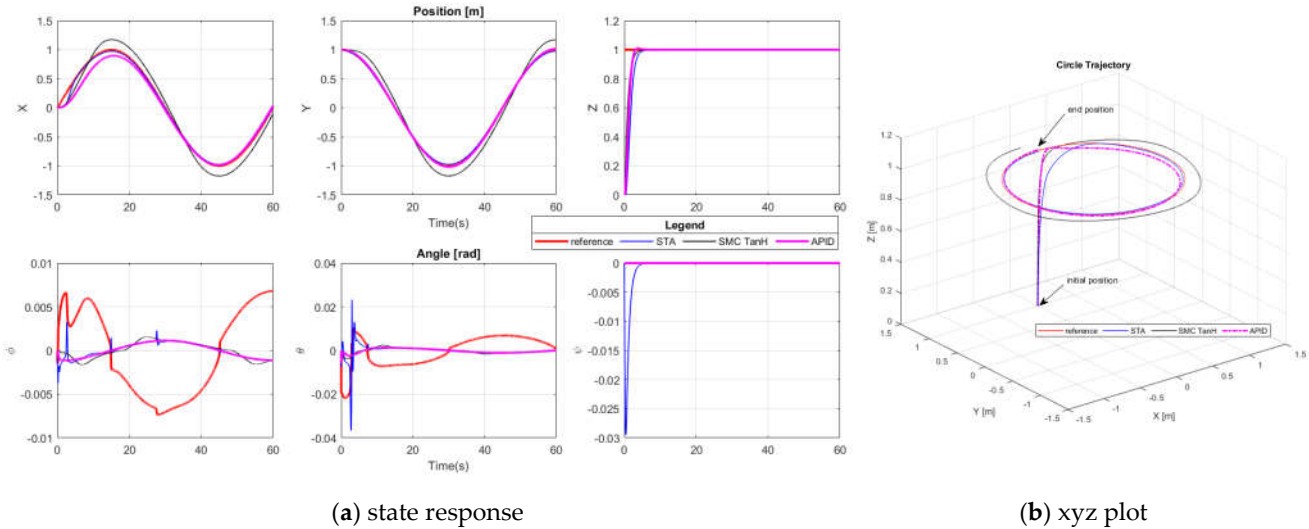

(**a**) state response　　　　　　　　　　　　　　　　(**b**) xyz plot

**Figure 4.** Circle trajectory without perturbation.

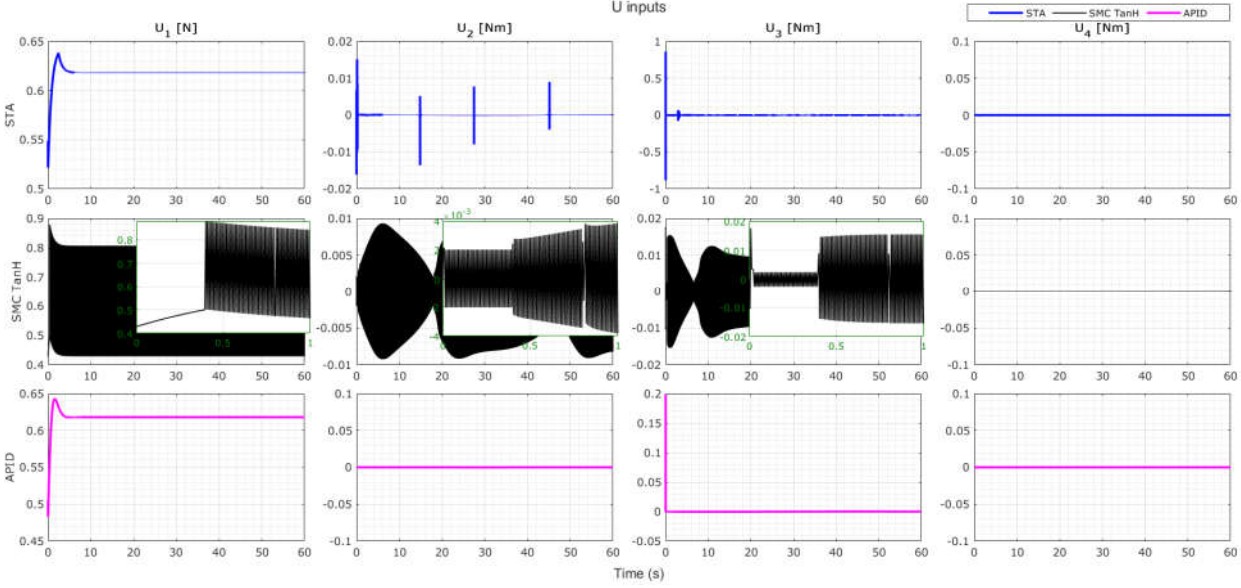

**Figure 5.** $U_1, U_2, U_3, U_4$ input for STA, SMC TanH, and APID controllers without perturbation.

### 4.2. Circle Trajectory with Perturbation

The simulation continued with a perturbation of a pulse with an amplitude of 0.025 injected to mimic a small perturbation every 15 seconds to observe the robustness of the proposed control scheme on $x, y,$ and $z$. In addition, a normal Gaussian noise was injected as $d_k = \mathcal{N}(0, 0.08)$ for $(k = \phi, \theta, \psi)$ [28]. The perturbation was relatively small because the MAV was smaller than 1 kg. A large force will make the system unstable and incapable of performing as desired.

The stabilization of state variables is shown in Figure 6a. All controllers work properly to converge to the desired point. For the desired trajectory, as shown in Figure 6b, the APID shows a more significant response to follow the desired reference with satisfaction of integral square error (ISE) performance compared to another controller, as stated in Table 2. Figure 7 shows the input response towards the system to achieve stabilization; only $U_1$ shows a ripple of pulse every 15 s for all controllers. However, the other U inputs remain steady. Hence, this clearly shows the most excellent performance of APID compared to another controller.

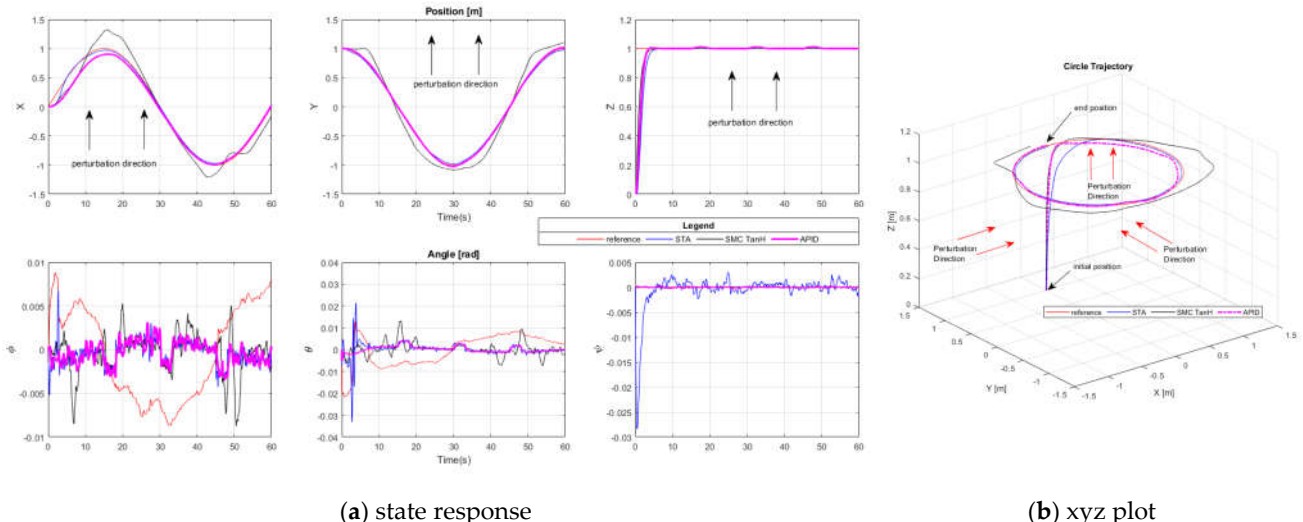

(**a**) state response                                                                 (**b**) xyz plot

**Figure 6.** Circle trajectory with perturbation.

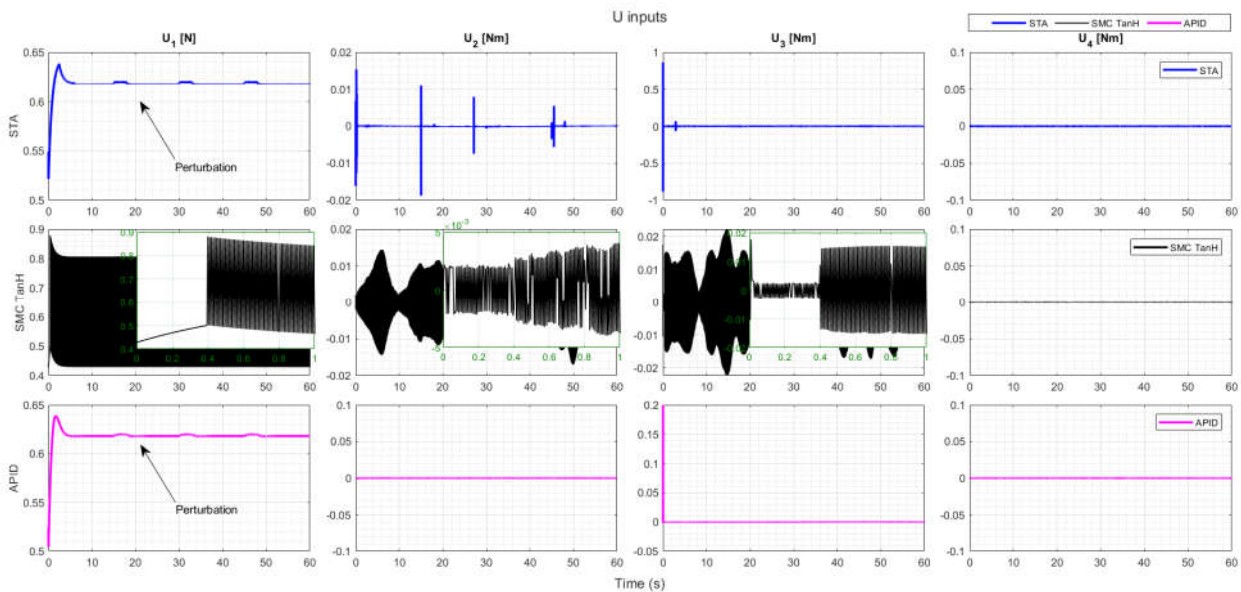

**Figure 7.** $U_1, U_2, U_3, U_4$ input for STA, SMC TanH, and APID controllers with perturbation.

### 4.3. Model-Based Simulation

The simulation continued using MATLAB/Simulink R2020b via a Simulink Support Package for Parrot Minidrones. This model is provided by MATLAB based on Aerospace Blockset developed by MIT [29]. This model-based simulation can perform hardware-in-loop, which shortens the programming process by directly using a Simulink block to perform the algorithm. This simulation is vital to show the effectiveness of the proposed control scheme before being implemented directly on a real-time platform.

The simulation starts by calling the 'parrotMinidroneHoverStart' command. The simulation used a third-order solver with a fixed sampling time of $Ts = 0.005$ seconds.

Table 3 shows the parameter used in the environment model. Based on sliding surface in (18), fuzzy logic adaptive parameter tuning law in (45), fuzzy logic compensator in (51), and adaptive parameter gains tuning law in (31), the control algorithm was designed by modifying the controller box inside the flight control system. The PID learning rate, fuzzy learning rate, and sliding parameter are listed in Table 4.

**Table 3.** Parameters in environment model.

| Parameter | Symbol | Constant Value |
|---|---|---|
| Air Temp | $T$ | 288 K |
| Speed of Sound | $a$ | 340 m/s |
| Pressure | $P$ | $101.3 \times 10^3$ Pa |
| Air Density | $\rho$ | 1.1840 kg/m$^3$ |

**Table 4.** APIDC learning rate, sliding parameter, and fuzzy learning rate.

| | Position | | | Attitude | | |
|---|---|---|---|---|---|---|
| **Learning Rate** | $x$ | $y$ | $z$ | $\phi$ | $\theta$ | $\psi$ |
| $\beta_p$ | 0.2 | 0.2 | 0.01 | 0.1 | 0.1 | 4 |
| $\beta_i$ | 0.7 | 0.7 | 0.01 | 0.1 | 0.01 | 0.01 |
| $\beta_d$ | 2 | 2 | 0.01 | 0.01 | 0.01 | 0.01 |
| Sliding constant | $x$ | $y$ | $z$ | $\phi$ | $\theta$ | $\psi$ |
| $k_1$ | 1 | 1 | 0.1 | 5 | 2 | 0.5 |
| $k_2$ | 0.01 | 0.01 | 0.5 | 2 | 1 | 2 |
| Fuzzy learning rate | $x$ | $y$ | $z$ | $\phi$ | $\theta$ | $\psi$ |
| $\eta_r$ | $-0.15$ | $-0.25$ | $-0.001$ | $-0.001$ | $-0.001$ | $-0.001$ |

The simulations are based on planning two paths, 'Orbit Follower' and 'Waypoint Follower', taken from the UAV toolbox. In addition, the landing logic (LL) is also implemented to prevent the Parrot Mambo Minidrone from crashing during landing. This LL will be triggered 5 seconds before stop times. The performance index for all states is recorder and tabulated in Table 5.

**Table 5.** Simulation performance index.

| | Position | | | Attitude | | |
|---|---|---|---|---|---|---|
| **IAE** | $x$ | $y$ | $z$ | $\phi$ | $\theta$ | $\psi$ |
| Orbit | 6.4330 | 6.2010 | 1.1290 | 0.03619 | 0.04502 | 0.00118 |
| Waypoint | 6.6441 | 4.8270 | 1.1280 | 0.05802 | 0.05134 | 0.00116 |

4.3.1. Orbit Follower Simulation

For this orbit follower, the orbit radius was set as 0.5 m, and in the clockwise orbit direction, LookaheadDistance was set at a minimum of 0.1 m to 0.195 m. The LookaheadDistance input was a positive scalar to control how closely the UAV follows the circular path. Small values improve tracking but can lead to oscillations in the path [30].

Figure 8 shows the response of the position and attitude states over 60 seconds of simulation. The figure indicates that the Parrot Mambo Minidrone can perform flight simulation using APIDC where the MAV can stabilize the altitude at 1 m height from the ground within 6 seconds of settling time, *Ts*, with slight overshoot. The roll, pitch, and yaw responses show it can track the produced reference signal. The 3-dimensional plot of $x$, $y$, and $z$ in Figure 9 clearly shows the robustness of the APIDC to reject external disturbances from the environment model and noise from the sensors model.

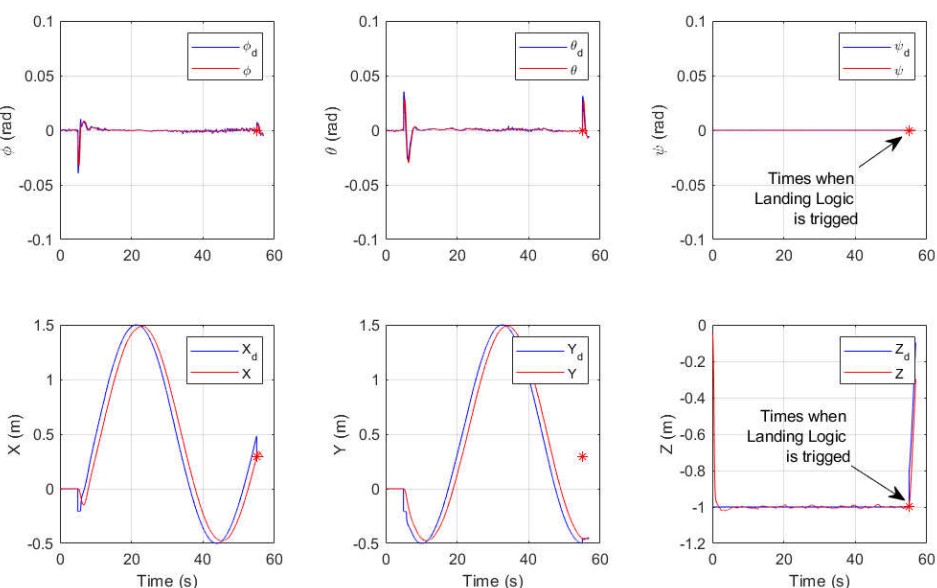

**Figure 8.** Orbit follower—position and attitude responses through simulation.

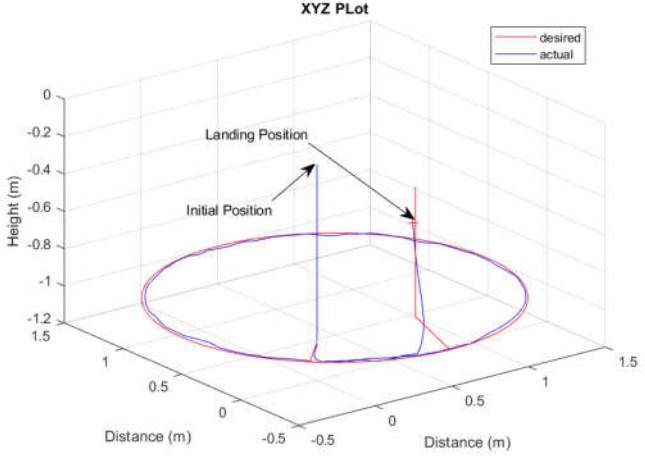

**Figure 9.** Orbit follower through simulation.

### 4.3.2. Waypoint Follower Simulation

For this waypoint follower, a square trajectory point was set as 1.5 m × 1.5 m above 1 m height. The LookaheadDistance was set at a minimum of 0.1 m to 0.2 m. The LookaheadDistance input is a positive scalar that controls how closely the UAV follows the path. Small values improve tracking but can lead to oscillations in the path [30].

The APIDC parameters were unchanged, as stated in Table 4. Figure 10 shows the position and attitude responses over 60 seconds of simulation. The figure indicates that the Parrot Mambo Minidrone is able to perform flight simulation using APIDC where the MAV can stabilize the altitude at 1 m height from the ground within 6 seconds of settling time, *Ts*, with slight overshoot. Furthermore, the roll, pitch, and yaw responses show it can track the produced reference signal. This shows the robustness of the APIDC to reject external disturbances from the environment model and noise from the sensors model. Figure 11 shows the *xyz* plot to describe the Parrot Mambo Minidrone movement based on the setup waypoint.

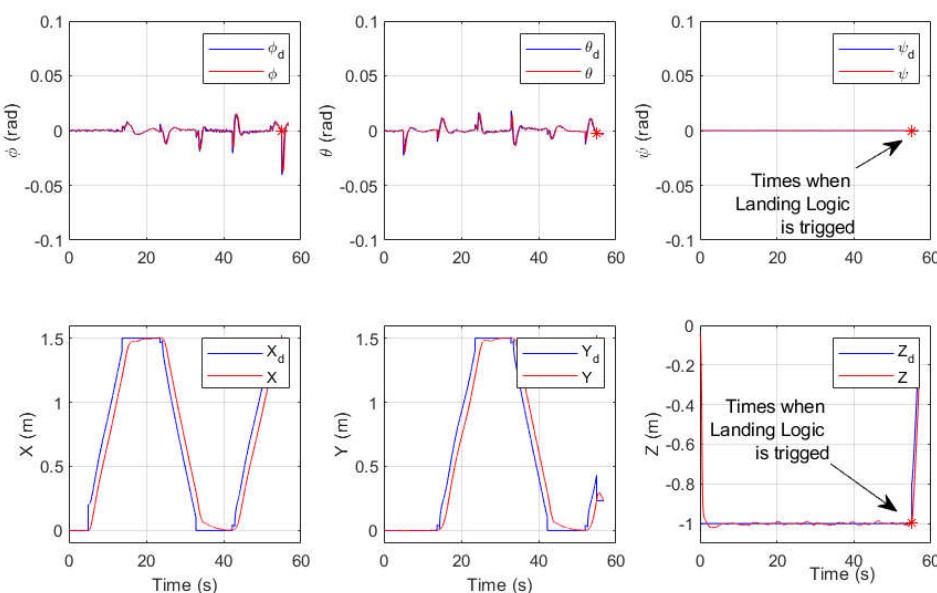

**Figure 10.** Waypoint follower—position and attitude response through simulation.

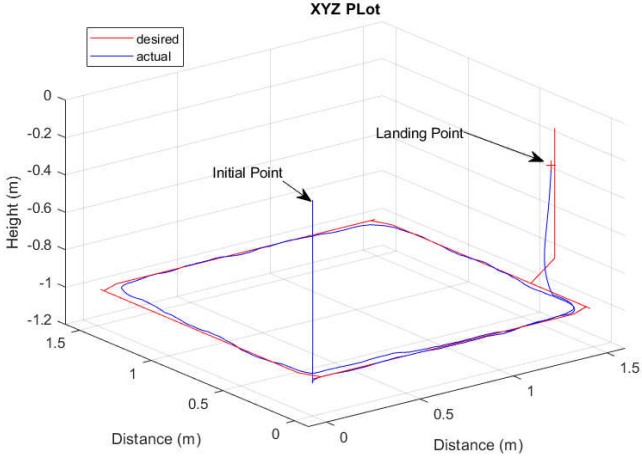

**Figure 11.** Waypoint follower through simulation.

## 5. Conclusions

In this paper, the simulation of SMC with STA, SMC with TanH, and Adaptive PID was provided and tested on the MAV quadrotor model. The control algorithms were tested without and with perturbation injected into the systems. After that, the adaptive PID was tested on model-based Parrot Mambo Minidrones provided by the MATLAB Simulink support package. This adaptive mechanism control, which was based on a second-order sliding mode, was used to tune the parameter gains of the proportional-integral-derivative controller. A fuzzy compensator was also used to reduce the chattering phenomena caused by the sliding mode control. Moreover, the automated tuning was based on the technique of gradient descent, and the Lyapunov stability theorem proved the stability of the proposed APIDC. Finally, results from the orbit follower and the waypoint follower demonstrate the advantages of the control algorithms, which produced better transient performance and faster convergence towards stability. In the future, a real-time experiment can be conducted to verify the above-proposed control scheme.

**Author Contributions:** Conceptualization, A.N. and M.A.M.B.; methodology, A.N.; software A.N.; validation, A.N. and M.A.M.B.; formal analysis, A.N.; investigation A.N., writing—original draft

preparation, A.N.; writing—review and editing, M.A.M.B. and Z.M.; supervision, M.A.M.B. and Z.M. All authors have read and agreed to the published version of the manuscript.

**Funding:** This research received no external funding.

**Data Availability Statement:** The study did not report any data.

**Acknowledgments:** The authors would like to thank Universiti Teknikal Malaysia Melaka (UTeM), Universiti Teknologi Malaysia (UTM), Ministry of Higher Education (MOHE) through the Fundamental Research Grant Scheme (FRGS/1/2021/TK0/UTM/02/56), and Advanced Academia-Industrial Testing Laboratory (AiTL), UTeM for supporting this research.

**Conflicts of Interest:** The authors declare no conflict of interest.

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
