# Peer review of "Position and Attitude Tracking of MAV Quadrotor Using SMC-Based Adaptive PID Controller"

_drones, doi:10.3390/drones6090263_

Round 1
Reviewer 1 Report
The article having titled as "Position and Attitude Tracking of MAV Quadrotor using SMC- 1 Based Adaptive PID Controller", lie in my area of expertise that why I suggest the follwoing points which may help the authors in order to imrpove the quality of this manuscript.
1. Abstract is written weak moreover no novelty in it
2. Introduction have many grammar isseus as well as authors donot cite the references properly
3. Need to add another heading after introduction that must be "State of the Art Work", and cite the following articles;
a. Ali, Zain Anwar, Daobo Wang, and Muhammad Aamir. "Fuzzy-based hybrid control algorithm for the stabilization of a tri-rotor UAV." Sensors 16, no. 5 (2016): 652.
b. Guo, Yufei, Leru Luo, and Changchun Bao. "Design of a Fixed-Wing UAV Controller Combined Fuzzy Adaptive Method and Sliding Mode Control." Mathematical Problems in Engineering 2022 (2022).
c. Yan, Li, Julian L. Webber, Abolfazl Mehbodniya, Balakrishna Moorthy, S. Sivamani, Shah Nazir, and Mohammad Shabaz. "Distributed optimization of heterogeneous UAV cluster PID controller based on machine learning." Computers and Electrical Engineering 101 (2022): 108059.
4. In all Equations must add brackets in the end
5. In position control heading (2.2) where is the altitude equations?
6. Check equation (22) carefully
7. In equation what about altitude which is shown in the trajectory
8. Delete Figure 8. Parrot Mambo Minidrone Flight Simulator Model by MATLAB.
9. Need to maximize the results
Summary: What is the ain motivation of this research and what is the problem statement and proposed solution its best to add in the article
Author Response
A response to the reviewer is attached. Please have a look at it.

Reviewer 2 Report
This manuscript introduces a method of multi-rotor aerial vehicle control that modifies a sliding mode algorithm for control. The manuscript presents the problem of disturbance rejection for the control problem of micro air vehicles, develops the dynamic equations of motions, develops the control strategy, and then provides simulation results comparing the proposed algorithm to that of others. The manuscript is relatively well written, although the authors might benefit from review by a native English speaker for grammatical corrections. I noticed some instances of missing or mixed articles (a vs. the). I do have some concerns about the level of professionalism in the writing with regards to image quality and units. Overall, the control strategy appears sound and the simulation result support the authors thesis of improved performance.
Primary Critique:
The proposed methods seem to be based on the work of two papers listed as references [21] and [22] in the manuscript. I commend the authors for properly attributing the work of those references, however the number of references to [21] and [22] make it difficult for me to distinguish what is new about the proposed algorithm. I believe that the novelty resides in the development of the fuzzy compensator and the associated stability analysis thereafter presented (sections 3.2 and 3.3). The authors would do well to clearly delineate in both the body of the manuscript and in the abstract what their contribution is to this field and how it is different from [21] and [22].
Secondary Critiques:
1. Why is the Simulink testing (section 4.3) necessary after the other simulation (sections 4.1 and 4.2). Why was it necessary to do all these simulations (rather than just one) and what did each result tell you that the other simulations did not. There should be a discussion section to summarize the simulation results for the reader.
2. The algorithm in lines 317-323 should follow the standard algorithm format. The authors may want to check the ‘algorithms’ Latex package. Also, it may help the reader to move this algorithm up to the beginning of section 3 to help provide a roadmap to the reader.
3. The authors state that the fuzzy compensator reduces error (see lines 213-215) but there is no justification for this statement or proof as to why this is the case. Can you run the simulations with and without the compensator to highlight the difference?
4. The values of ‘r’ in equation 33 are never discussed but are the primary focus of section 3.2 and 3.3. What are they – simply weights on the firing strengths (w_i)? Also, what do you mean by ‘firing strength’. Please clarify the variables of section 3.2.
5. On lines 327-328 you introduce other algorithms that you will compare against. Please provide citations for these.
6. Lines 336-343. Just listing out all the parameters without discussing how you arrived at them is inadequate. Please justify the selections. Given the number of parameters, it is not clear that the simulation results you provide actually proves your algorithm improves performance as compared to the other algorithms. Might some other combination of parameters make you algorithm perform worse than the others? Please use this though to help justify your parameter selection and convey that justification to the reader so that we can be assured this is a ‘fair’ comparison.
7. The figures in the simulation section are all too small. Fix the size for the axis labels and figure sizes so they are readable on a printed page. The simulation section plots look like those I get from undergraduate student lab reports. These aren’t yet fit for publication. I know that blob of black ink in the SMC TanH results is because of oscillation, but you need to change the line width so this is clear to the reader.
8. Why is there no caption for table two? Also, there are no units in this table. Once again, this looks like an undergrad report.
9. The text in figure 8 is too small to read. Fix this.
10. Units in table 5?
Author Response

(The authors gave the same response as above.)

Reviewer 3 Report
This manuscript has primarily concerned with position and attitude tracking of quadrotor using Adaptive PID Controller. In particular, an adaptive mechanism base on second-order sliding mode control is used to tune the parameter gains and a fuzzy compensator is proposed to reduce the chattering phenomena. The authors have a good grasp of the background of application problem, while the corresponding solutions are proposed to handle the related issues. Some simulation experiments are also conducted to verify the conclusion. This manuscript contains interesting results but in my opinion, its novelty is marginal such that the current form is not qualified for publication in drones. The following issues in the manuscript need to be improved/modified. 1). The authors have not extracted the really valuable highlights in this manuscript, such that it is hard to arouse the readers' interest. 2). It is better to provide some real experiments of quadrotor control to further verify the performance of the proposed method, only simulation experiments are conducted and it is not convincing. Or the authors may explain the reason why the real experiments are not performed.3). Some equation expressions should be strictly written and corrected. For example, Equation (23) and (31), the symbol sgn should be written as sign? And in the line 169, there is a format error.
Author Response

(The authors gave the same response as above.)

Round 2
Reviewer 1 Report
I recommend to accept the article in its present form. The authors revised the article as per the given suggestions by me.